# Natural hyperbolicity of hexagonal boron nitride in the deep ultraviolet

Bongjun Choi [1], Jason Lynch [1], Wangleong Chen[2], Seong-Joon Jeon[3,4], Hyungseob Cho[3,4], Kyungmin Yang[1], Jonghwan Kim [3,4,5] ✉, Nader Engheta[1,2,6,7] ✉ & Deep Jariwala [1,2] ✉

Hyperbolic media enable unique optical phenomena including hyperlensing, negative refraction, enhanced photonic density of states (PDOS), and highly confined polaritons. While most hyperbolic media are artificially engineered metamaterials, certain natural materials with extreme anisotropy can exhibit hyperbolic dispersion. Here, based on experimental evidence and theoretical fitting estimates to the experimental data, we suggest the presence of natural hyperbolic dispersion in hexagonal boron nitride (hBN) in the deep-ultraviolet (DUV) regime, induced by strong, anisotropic exciton resonances. Using all-optical imaging spectroscopic ellipsometry (ISE), we characterize the complex dielectric function along in-plane and out-of-plane directions down to 190 nm (6.53 eV), revealing a potential type-II hyperbolic window in the DUV regime. We predict that hyperbolicity supports hyperbolic exciton polaritons (HEP) with high directionality and slow group velocity, as confirmed by numerical calculations. Our findings suggest hBN as a platform for nanophotonic applications in the technologically significant DUV spectral range.

Nanoscale light manipulation is crucial for developing next-generation nano-optical devices for sensing[1], waveguiding[2], and imaging[3]. Hyperbolic media, characterized by permittivities of opposite sign along different crystal axes (e.g., $\mathrm{Re}(\varepsilon_{\parallel}) \times \mathrm{Re}(\varepsilon_{\perp}) < 0$), have attracted significant attention due to their ability to support large wavevectors, enabling extreme sub-diffraction light confinement, and large photonic density of states (PDOS)[4,5]. Since hyperbolic dispersion requires large anisotropy, it is typically achieved in artificially made hyperbolic metamaterials (HMMs)[3,6–9], which are realized by subwavelength multilayer stacks of alternating dielectric and metallic films, exhibiting hyperbolic dispersion because it heavily relies on the plasmonic resonance of the metal layers[4]. These HMMs exhibit an open isofrequency surface, supporting the propagation of high-momentum waves and enabling unique phenomena like negative refraction[10], sub-

diffraction imaging via hyperlensing[3,8,9], and substantial enhancement of the Purcell factor[11].

Van der Waals (vdW) two-dimensional (2D) materials exhibit inherent high anisotropy along the out-of-plane direction due to their layered structure. This pronounced anisotropy is a key prerequisite for hyperbolic dispersion[12–14]. Moreover, the pronounced anisotropy of certain vdW 2D crystals gives rise to strongly direction-dependent resonances–phonons, plasmons, and excitons–which profoundly impact their dielectric response[15]. With the discovery of vdW 2D crystals, hyperbolic dispersion can be more readily achieved, rather than relying on plasmonic multilayer HMMs. Prior results have revealed hyperbolic behavior originating from the anisotropic resonances of phonons in $\alpha$-$MoO_3$[16] and hexagonal boron nitride (hBN)[17], plasmons in $WTe_2$[18], and excitons in monolayer black phosphorus

¹Department of Electrical and Systems Engineering, University of Pennsylvania, Philadelphia, PA, USA. ²Department of Materials Science and Engineering, University of Pennsylvania, Philadelphia, PA, USA. ³Department of Materials Science and Engineering, Pohang University of Science and Technology, Pohang, Republic of Korea. ⁴Center for van der Waals Quantum Solids, Institute for Basic Science (IBS), Pohang, Republic of Korea. ⁵Department of Physics, Pohang University of Science and Technology, Pohang, Republic of Korea. ⁶Department of Bioengineering, University of Pennsylvania, Philadelphia, PA, USA. ⁷Department of Physics and Astronomy, University of Pennsylvania, Philadelphia, PA, USA. ✉e-mail: jonghwankim@postech.ac.kr; engheta@seas.upenn.edu; dmj@seas.upenn.edu

(BP)[19] and chromium sulfide bromide (CrSBr)[20] at cryogenic temperatures, as well as in various 2D vdW heterostructures[2,21–23].

Natural hyperbolic media support hybrid electromagnetic modes (hyperbolic polaritons) that enable broadband, deeply subdiffractional confinement with strongly directional propagation. Natural hyperbolic phonon polaritons were first observed in hBN crystals experimentally by near-field measurements in the mid-infrared (MIR)[24]. Since then, various hyperbolic polaritons driven by phonons[16,24], plasmons[25], and excitons[20] have been demonstrated across many material platforms. While phonon- and plasmon-mediated hyperbolicity has been extensively explored, exciton-induced hyperbolicity remains relatively rare, as it requires materials with exceptionally strong excitonic resonances with extreme anisotropy.

hBN is a well-known vdW polar dielectric exhibiting hyperbolic dispersion in the MIR regime due to anisotropic phonon resonances[15,24]. It has also attracted attention as an excitonic material due to its strong emission in the deep-ultraviolet (DUV) regime (around 180 ~ 300 nm, 4.13 ~ 6.89 eV)[26]. Even though the fundamental bandgap of hBN is an indirect bandgap (K → M point) in the DUV regime, as proved by the two-photon excitation spectroscopy[27], it also has a direct transition near ~ 6.1 eV at the K valley, supported by numerous density-functional theory (DFT) calculations and experiments[28–30]. Strong in-plane covalent bonding and weak out-of-plane van der Waals interactions create tightly bound 2D excitons with the large binding energy, $E_b$ ~ 700 meV, for the lowest exciton[28,31–33]. In addition, the electronic flatband of hBN in the Brillouin zone facilitates strong light-matter interaction[28].

We suggest the presence of natural hyperbolic dispersion in hBN, arising from its anisotropic strong exciton resonances in the DUV regime. Here, using imaging spectroscopic ellipsometry (ISE), we characterize the complex refractive index and exciton resonance properties of hBN along in-plane (∥) and out-of-plane (⊥) directions. hBN shows pronounced exciton oscillator strength in the in-plane direction compared to the out-of-plane direction, suggesting the presence of a hyperbolic window in the DUV regime. We predict that this hyperbolicity enables a transition in the isofrequency contour from elliptic (closed) to hyperbolic (open) shape[5], supporting hyperbolic exciton polaritons (HEP) with high directionality and confinement[15,20,34]. Our findings would enable light manipulation at the nanoscale in the DUV regime, which is critical for extreme-resolution photolithography using 193 nm ArF excimer lasers and DUV nonlinear optics.

## Results and Discussion

While hBN's optical properties in the DUV have been widely studied, most prior studies have focused on its electronic band structure calculation from the ab initio approach and its interband transitions by inelastic electron scattering (IES) measurement[28,30,32,35]. In contrast, spectroscopic ellipsometry studies, which provide direct and sensitive optical characterization in the DUV regime, remain scarce[14,36]. Particularly, refractive indices resolved separately along the in-plane and out-of-plane directions are rare, yet they provide indispensable insight into the material's anisotropy and hyperbolic nature in the DUV excitonic regime. Recent ellipsometry studies have revealed a high refractive index and strong birefringence of hBN in the UV regime (around 250 nm), although they did not capture the excitonic resonance in the DUV regime[14]. We first characterize the excitonic features and refractive index of hBN along in-plane (∥) and out-of-plane (⊥) directions using ISE down to the DUV to NIR regimes (190 ~ 1000 nm, 1.24 ~ 6.53 eV). First, we prepare the hBN sample using mechanical exfoliation with tape and transfer it onto the target substrate (See Methods). We chose c-plane $Al_2O_3$ as the substrate, which has a wide bandgap to increase the reflectance in the DUV regime. Since the excitonic resonances of hBN occur in the DUV range, where most materials (e.g., Si) strongly absorb light, using an $Al_2O_3$ substrate minimizes absorption and enhances the reflected light intensity (Supplementary Fig. 1). When linearly polarized light is incident on a sample, strong excitonic resonances in its dielectric function cause variations in the Fresnel reflection coefficients $r_p$ and $r_s$, consequently changing the ellipsometry parameters $\Psi$ (the amplitude ratio) and $\Delta$ (the phase difference) with high sensitivity, enabling exciton behavior characterization[37]. Furthermore, obliquely incident light distinguishes a sample's electromagnetic response parallel to the surface from that perpendicular to it (Fig. 1a)[37]. As shown in Fig. 1b, an exfoliated hBN flake was characterized by ISE. The ~ 1 μm lateral resolution of our ISE allows detailed optical properties ($\Psi$ and $\Delta$) map data to be acquired as a function of incidence angle and wavelength on the small flakes that are inaccessible with conventional spectroscopic ellipsometry. The Raman spectrum of hBN exhibits a sharp peak at 1370 cm⁻¹, the characteristic $E_{2g}$ vibrational mode, corresponding to the in-plane stretching of B and N atoms, as illustrated in the inset (Fig. 1c)[31]. The narrow peak with a full width at half maximum (FWHM) of approximately 9.2 cm⁻¹ ( ~ 8 cm⁻¹ for flux grown and ~30 cm⁻¹ for MOCVD) indicates that the hBN possesses high crystalline quality with minimal structural disorder and low strain, both of which are essential for sustaining strong excitonic resonances[38,39]. We obtain the $\Psi$ and $\Delta$ (Supplementary Fig. 2) map and make the optical model using multiple Lorentz oscillator models to capture the exciton resonance behavior in hBN (see Supplementary Note 1)[37,40–45]. We use phenomenologically extracted parameters from ellipsometry because they provide a convenient and reliable basis for the Lorentz model, avoiding the need for direct measurements of radiative and non-radiative rates. Because hBN is an uniaxial material, we modeled the in-plane and out-of-plane dielectric functions independently using distinct multiple Lorentz oscillator models. To improve the reliability of the results, multiple hBN thicknesses are measured at various incidence angles between 40° and 70° with respect to the sample's surface. The large variation in incidence angle, and thick flakes, yields a strong out-of-plane signal allowing us to determine the uniaxial dielectric function with high fidelity. The multiple Lorentz oscillator model captures the interband transitions in hBN along different optical axes with high accuracy, showing strong agreement with the experimentally obtained $\Psi$ and $\Delta$ values across various incident angles and hBN thicknesses (Supplementary Fig. 3).

The measured hBN's complex refractive index is shown in Fig. 1d along the in-plane (∥) and out-of-plane (⊥) directions. In the in-plane direction, hBN exhibits a pronounced excitonic resonance near 6.14 eV, characterized by a large extinction coefficient (k), which arises from the direct π → π* transition at the K point in the reciprocal space. This observation aligns well with the theoretical electronic band calculation using the ab initio (GW + BSE) method[28,32] as well as prior experimental measurements using synchrotron radiation[36] and IES[30]. Although bulk hBN possesses an indirect bandgap, phonon-assisted (indirect) optical transitions are essentially absent from the measured spectra since their oscillator strength is orders of magnitude weaker than that of the dominant excitonic resonance[46,47]. Furthermore, hBN exhibits a high refractive index ( ~ 2.5 at 250 nm) in the UV regime due to strong excitonic resonances. These characteristics make hBN an attractive high-index material alternative to conventional low-index UV materials, such as fused silica ($SiO_2$, $n$ ~ 1.5 at 250 nm)[48] and magnesium fluoride ($MgF_2$, $n$ ~ 1.4 at 250 nm)[49]. On the other hand, hBN exhibits the interband transition along the out-of-plane direction approximately at 6.05 eV with a small extinction coefficient (k) compared to that of the in-plane components due to its highly in-plane oriented exciton nature[31]. This observation is well aligned with the previous literature[30], manifesting strong and anisotropic excitonic behavior, which induces hyperbolic dispersion, discussed later. As shown in Fig. 1e, we plot the complex permittivity ($\varepsilon = \varepsilon_1 + i\varepsilon_2$) of hBN, where $\varepsilon_1 = n^2 - k^2$ and $\varepsilon_2 = 2nk$, along both its in-plane and out-of-plane directions. The

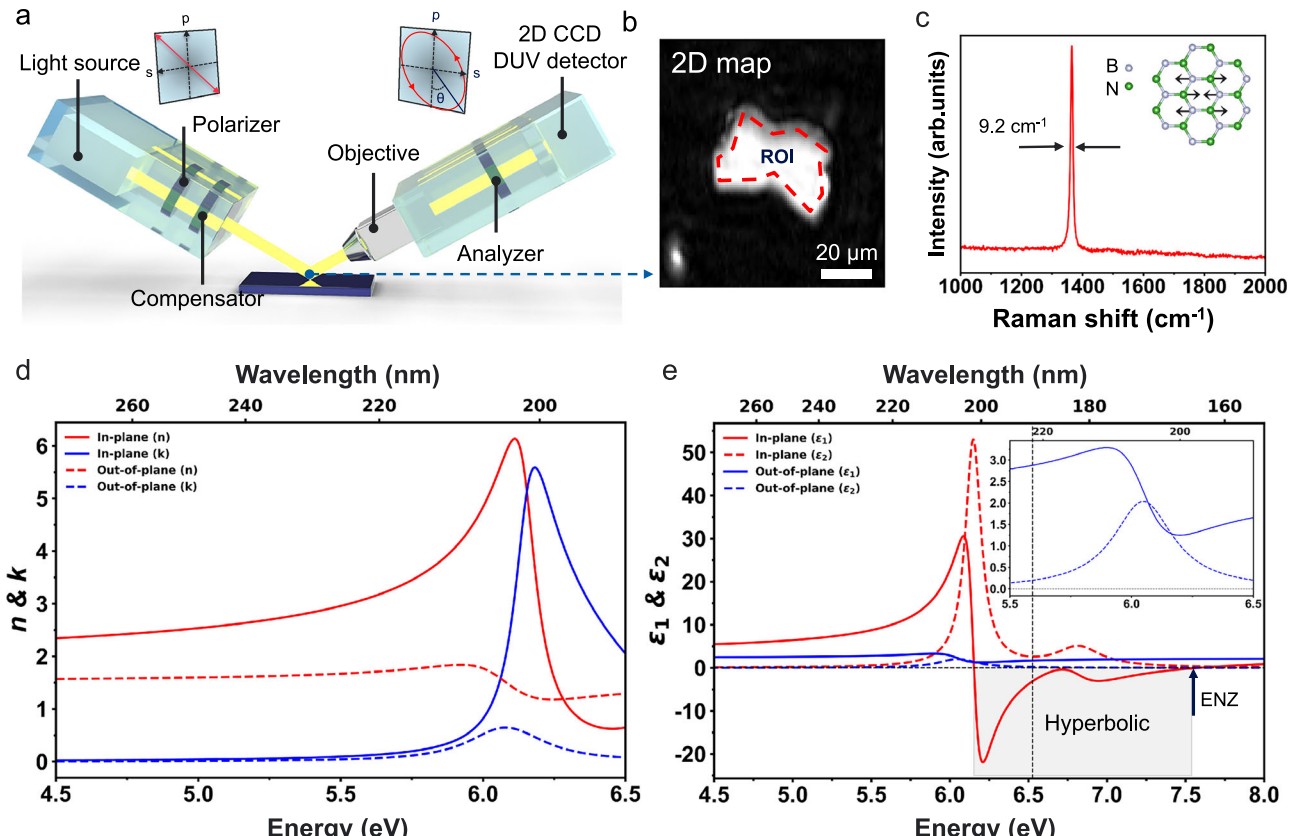

**Fig. 1 | Imaging spectroscopic ellipsometry measurement of hexagonal boron nitride (hBN). a** Schematic illustration of imaging spectroscopic ellipsometry (ISE). Unlike conventional spectroscopic ellipsometry, imaging ellipsometry uses a high-resolution (∼1 μm) 2D charge-coupled device (CCD) detector with an objective lens (7×) to map Ψ and Δ values, allowing the extraction of optical properties from even small flakes. **b** Example of the mapped image data of a small hBN flake using ISE. The red dashed outline indicates the region of interest (ROI). **c** Raman spectrum obtained from measured hBN shows a clear $E_{2g}$ resonance with a narrow full width at half maximum (FWHM), demonstrating the high quality of the measured hBN. The inset shows a schematic illustration of the in-plane stretching mode of B and N atoms. **d** Refractive index (n) and extinction coefficient (k) of hBN along the different optical axes (in-plane (∥) and out-of-plane (⊥)) showing strong excitonic

features of hBN. These extracted values from ISE are in agreement with prior measurements using synchrotron radiation and inelastic electron scattering[30,36]. **e** Complex permittivities ($\varepsilon = \varepsilon_1 + i\varepsilon_2$) of hBN along different optical axes (in-plane (∥) and out-of-plane (⊥)) indicating apparent hyperbolic dispersion (gray shaded area) in the deep-ultraviolet (DUV) regime with epsilon-near-zero (ENZ) point. The permittivities are extrapolated based on the multiple Lorentz oscillator model obtained from the ISE measurement above (below) 6.53 eV (190 nm). The dashed vertical line represents the boundary between the experimental and extrapolated regimes. The inset presents a magnified view of the out-of-plane complex permittivity in the excitonic regime near 6.05 eV. All complex refractive indices are listed in the Supplementary Table 1.

permittivity was directly retrieved from ISE measurements down to 190 nm (6.53 eV), a measurable wavelength regime from our tool. To extend our model to shorter wavelengths (∼155 nm, ∼8.0 eV), we extrapolated the data using a multi Lorentz oscillator model (See Supplementary Note 1) obtained from multiple samples and incidence angles. Since the higher energy electron transition above 6.53 eV is out of range from our tool, we adapted the previous synchrotron study to capture the high energy interband transitions (e.g., π → π* transition (∼6.14 and 6.82 eV) and σ → σ* transition (∼15 eV))[36]. Due to its pronounced anisotropic excitonic resonances, we suggest that hBN supports a well-defined DUV type-II hyperbolic window around 6.17–7.56 eV (164–201 nm), where $\varepsilon_{1,\parallel} < 0$ and $\varepsilon_{1,\perp} > 0$. We further identify an intrinsic epsilon-near-zero (ENZ) wavelength in hBN near 7.56 eV, where the losses are minimal. These naturally occurring hyperbolicity and ENZ points promise new opportunities for DUV nanophotonic device engineering and quantum-optical applications[50,51].

Since excitonic resonances play a critical role in determining the optical properties—such as the dielectric functions and potential hyperbolic dispersion—of 2D excitonic materials, we investigate the excitonic feature of hBN to analyze the origin of hyperbolicity by evaluating its exciton oscillator strength from the interband oscillator

strength in ellipsometry measurements (Fig. 2a). Generally, because 2D materials strongly support confined excitons in two dimensions with reduced dielectric screening, they exhibit significantly larger $E_b$ and strong oscillator strengths compared to their bulk counterparts[52]. The exciton oscillator strengths ($f_{exc}$) can be calculated using the simple formulation proposed by Ishihara et al.[53].

$$f_{exc} = \frac{2fS_0}{\pi a_{2D}^2} \ or \ \frac{2fV_0}{\pi a_{3D}^2} \quad (1)$$

where $f$ is the interband oscillator strength, $S_0 (V_0)$ is a unit-cell area (volume), and $a_{2D(3D)}$ is the material's 2D (3D) Bohr radius. Since a smaller exciton Bohr radius is advantageous for the strong oscillator strength, $f_{exc}$ shows an inversely square (cubic) proportional relation to Bohr radius, respectively. Tightly bound hBN excitons exhibit a small Bohr radius[31], resulting in a pronounced $f_{exc}$ and $E_b$ compared to the well-known highly excitonic materials, such as monolayer transition metal dichalcogenides (TMDCs), and 2D Ruddlesden-Popper (RP) perovskites. Monolayer TMDCs exhibit a large oscillator strength, although it is slightly smaller than that of hBN. The oscillator strength can increase at low temperatures, where excitons become more stable and nonradiative broadening is suppressed. Consequently, several

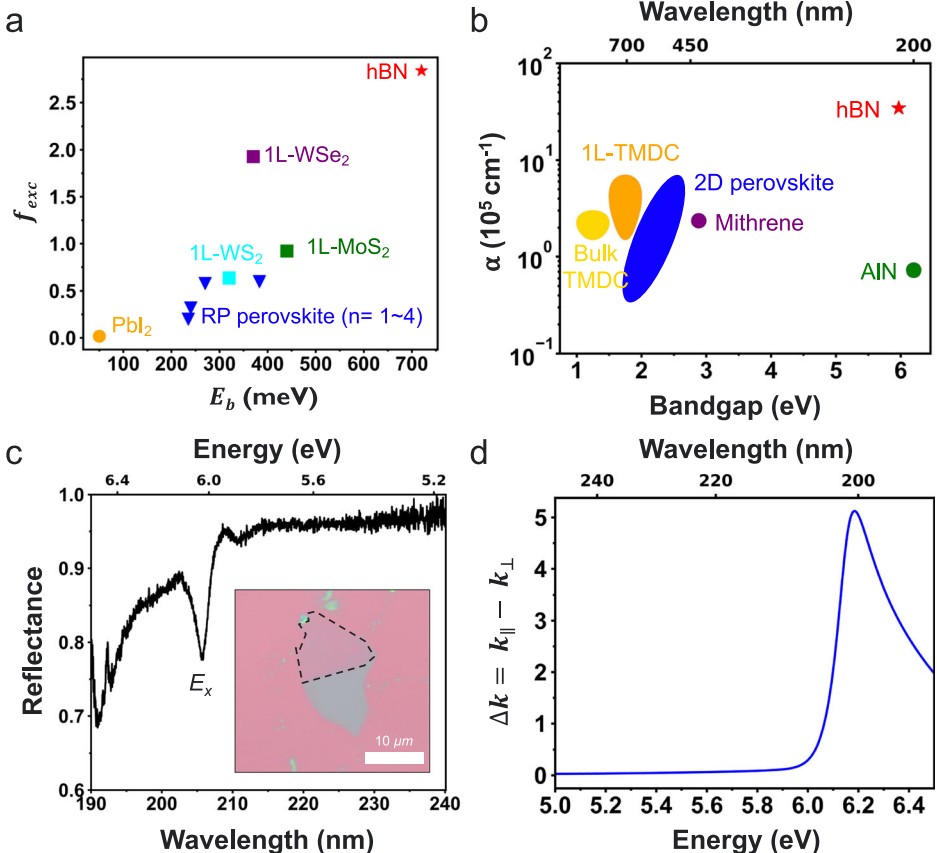

**Fig. 2 | Exciton properties of hBN. a** Comparison of exciton oscillator strength and exciton binding energy of various excitonic materials (hBN[32], monolayer transition metal dichalcogenides (TMDCs)[19,71–73], 2D Ruddlesden-Popper (RP) perovskite[55,74], and PbI₂[53]), showing hBN's strong oscillator strength and exciton binding energy compared to the other excitonic materials. The exciton oscillator strengths are calculated using Eq. (1), and parameters, such as interband oscillator strength, Bohr radius, and $E_b$ are adapted from prior literature[19,32,53,55,71–74]. **b** Comparison of the absorption coefficient along the in-plane directions of various excitonic (hBN, Bulk and monolayer of TMDCs ($MoS_2$, $MoSe_2$, $MoTe_2$, $WSe_2$, and $WS_2$), 2D RP and Dion-Jacobson (DJ) perovskites, mithrene, and AlN)[55,75–79], illustrating significant absorption coefficient due to the tightly bound excitons in hBN. **c** Reflectance spectrum ($R_{hBN}/R_{Al}$) in a 5L-hBN/Al structure, demonstrating the high absorbance from the thin hBN layer due to strong exciton resonance. The Al substrate serves as a back reflector, enhancing the excitonic absorption. The reflectance dips around 190 nm (6.53 eV) are originated from strong absorption of air. Inset shows image of thin hBN flake on a Si/SiO₂ substrate before the pick-up transfer. Note that the thickness of hBN was determined by the ISE measurement (Supplementary Fig. 5) **d** Calculated linear dichroism ($\Delta k = k_\parallel - k_\perp$) as a function of photon energy, revealing a large dichroism originated from anisotropic exciton resonances.

studies have reported hyperbolic dispersion of 2D vdW materials (e.g., BP and CrSBr) at low temperatures, where excitonic resonances are enhanced, highlighting the critical role of excitons in governing hyperbolicity[19,20,54]. Furthermore, RP perovskites exhibit a near-hyperbolic dispersion around exciton resonance, characterized by Re($\varepsilon_\parallel$) approaching zero while $Re(\varepsilon_\perp)$ remains positive, suggesting that true hyperbolicity could potentially emerge at low temperatures[55]. For comparison, PbI₂, which does not support strong excitonic effects, exhibits small values of $f_{exc}$ and $E_b$, indicating that excitons play an important role in the material's optical properties[53]. Figure 2b shows the absorption coefficient, $\alpha = 4\pi k/\lambda$, of various well-known excitonic materials including hBN (Bulk and monolayer TMDCs, 2D RP and Dion-Jacobson (DJ) perovskites, mithrene (AgSePh), and AlN). The giant excitonic oscillator strength of hBN enables a strong absorption peak around the exciton energy approximately $3.5 \times 10^6 \ cm^{-1}$ (Supplementary Fig. 4), which is consistent with the observation in Fig. 2a. Because exciton oscillator strength scales with optical absorption, monolayer TMDCs, where reduced dielectric screening enhances excitonic features, such as $E_b$ and $f_{exc}$, exhibit more significant absorption coefficients than their bulk counterparts. Notably, hBN's tightly bound exciton gives rise to absorption coefficients nearly an order of magnitude higher than other excitonic semiconductors. Even

compared to wide-bandgap AlN, whose band edge lies similarly in the DUV, hBN's excitonic resonance boosts its absorption by approximately two orders of magnitude, underscoring the dramatic impact of its bound excitons. Therefore, excitons dominate the optical responses in the DUV regime of hBN.

To probe the strong excitonic response, we measure the reflectance spectrum of thin hBN (5 layers) on an Al substrate in the DUV regime. The spectrum reveals pronounced absorption near the exciton energy even from a very thin hBN layer, as shown in Fig. 2c. Notably, most of the observed absorption originates from the hBN layer rather than the underlying Al substrate, and the exciton resonance exhibits a slight shift from its intrinsic energy as a result of phase accumulation across the layer[56], as confirmed by the transfer matrix method (TMM)[57,58] calculation results (Supplementary Fig. 6). More importantly, these strong excitons only exist along the in-plane direction, resulting in a large dichroism ($\Delta k = k_\parallel - k_\perp$) around 5, which reflects the difference in absorption properties along different optical axes, a prerequisite for natural hyperbolicity. Along with dichroism, hBN shows large birefringence in the DUV regime since dichroism and birefringence are related under the Kramers-Kronig relation (Supplementary Fig. 7)[37]. In addition, the large birefringence of hBN, even well below the band gap where loss is low, makes it uniquely suited for UV

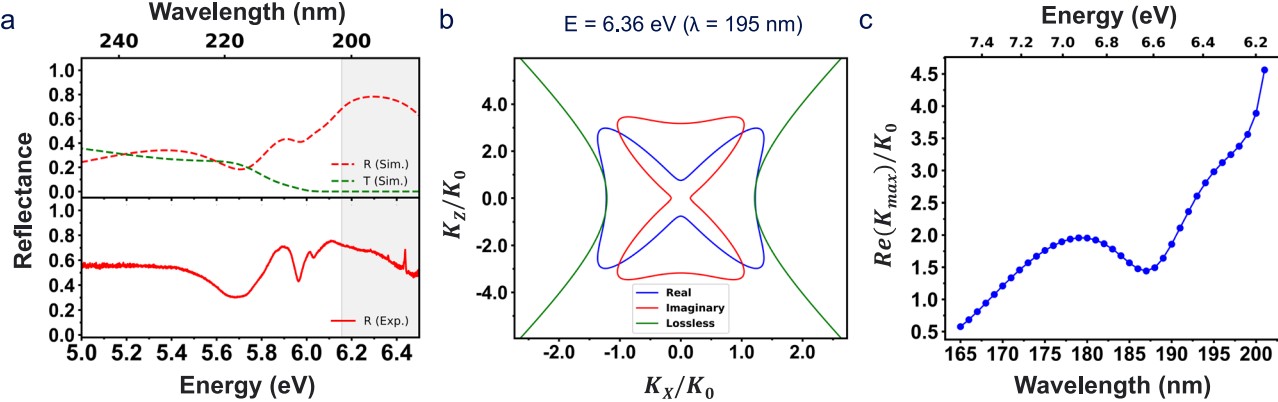

**Fig. 3 | Natural hyperbolic dispersion in hBN. a** Measured (solid red curve, bottom panel) and simulated (dotted red curve, top panel) reflectance, along with simulated transmittance (dotted green curve, top panel), from a 63 nm hBN sample on a c-plane Al₂O₃ substrate in the DUV regime, showing the highly reflective plateau (gray shaded area) associated with the hyperbolicity. **b** Isofrequency surface of hBN at 6.36 eV (195 nm) for transverse-magnetic (TM) plane wave with real loss (red and blue curves) and no loss (green curve), showing clear hyperbolic dispersion attributed to different signs of permittivities along different optical axes. **c** The max wavevector can be supported by the hBN as a function of wavelength (photon energy), reaching approximately 4.6 around the exciton resonance.

and DUV polarization optics; an attribute few other materials can match[14].

In its type-II hyperbolic regime (6.17–7.56 eV), hBN exhibits a metallic in-plane permittivity while maintaining a dielectric response along the out-of-plane direction, a characteristic feature of type-II hyperbolic materials, and shows a highly reflective plateau in the hyperbolic regime. Our measured and simulated reflectance spectrum from hBN (63 nm) on Al₂O₃ substrate exhibit the high reflectivity plateau of the hBN (gray-shaded region), a consequence of its hyperbolic dispersion (Fig. 3a)[17,28]. The highly reflective plateau is observed in hBN flakes of various thicknesses (see Supplementary Fig. 8a). Note that we chose an hBN thickness exceeding approximately 20 nm, where hBN exhibits highly reflective behavior due to coherent radiative decay, as confirmed by the TMM calculation shown in Supplementary Fig. 8b. As hBN approaches the optically thin limit, strong excitonic damping becomes dominant. This causes the broad plateau to evolve into a single resonance peak, consistent with the observation in Fig. 2c and analogous[4] to the phononic Reststrahlen band in the mid-infrared (MIR) regime of hBN[23,59]. In addition, the simulated transmission spectrum dips to near zero, confirming that electromagnetic waves cannot propagate normally through the hBN in this band except as evanescent/high-k modes as shown in Fig. 3a. The reflective plateau exhibits a highly reflective nature across a wide range of incident angles in the hyperbolic regime (Supplementary Fig. 8c).

The hyperbolic features are manifested only under transverse-magnetic (TM)-polarized light, as it probes both in-plane and out-of-plane dielectric responses. In contrast, transverse-electric (TE)-polarized light is sensitive only to the in-plane permittivity and may not exhibit hyperbolic behavior. For the TM plane wave, Maxwell's equation yields a dispersion relation: $\frac{k_x^2}{\varepsilon_{\perp}} + \frac{k_z^2}{\varepsilon_{\parallel}} = k_0^2$, where $k_0 = \frac{\omega}{c}$ is the free space wavevector, and $k_x$ and $k_z$ are the wavevectors in the x-plane and z-plane, respectively. Based on the dispersion relation, Fig. 3b shows a clear hyperbolic isofrequency surface of hBN at 6.36 eV (195 nm) for each lossy (red and blue) and lossless (green) case, while hBN exhibits an elliptical dispersion outside the hyperbolic region as expected (Supplementary Fig. 9). The anisotropic excitonic resonance suggests the possibility of a hyperbolic dispersion, supporting highly confined light and large PDOS, enabling the formation of hyperbolic polaritons discussed later[15,20,60]. In an ideal, lossless hyperbolic medium, the dispersion relation admits modes with infinitely large wavevectors in theory, formally yielding an infinite PDOS. In practice, however, material absorption introduces damping that imposes a finite cutoff on the maximum supported k-vector, resulting in a closed

isofrequency surface[61], as shown in Fig. 3b. Thus, it becomes essential to quantify the impact of absorption, especially in regimes where hyperbolic dispersion is driven by exciton resonances, since these same resonances impart strong intrinsic damping. Over hBN's hyperbolic band, we compute the largest wavevector the material can support as a function of photon energy (wavelength). Approaching the exciton resonance, hBN's hyperbolic bands extend to larger wavevectors, albeit with rising absorption. Supplementary Fig. 10 compares the loss-inclusive isofrequency contours at multiple photon energies (wavelength): as one approaches the exciton resonance (~6.14 eV), the real isofrequency surface expands (indicating larger wavevector) while the imaginary isofrequency surface grows simultaneously, indicating increased damping near the resonance. In its most extreme hyperbolic regime, hBN reaches $Re(K_{max})/K_0$ ~ 4.6 (Fig. 3c). Additionally, the enlarged wavevector space within hBN's hyperbolic band significantly enhances the PDOS, which is calculated by the Green's functions, resulting in an enhanced spontaneous emission rate, as quantified by the Purcell factor shown in the Supplementary Fig. 11[4,62].

We further investigate the highly confined HEP associated with the hyperbolic dispersion. The numerically simulated wave launched by a point source oriented along the z-axis, as illustrated in the inset of Fig. 4a, displays a circularly shaped radial wave propagation at 6.36 eV (195 nm) in the XY plane. Since radiation in the XY plane consists solely of the $E_z$ component, governed exclusively by a positive $\varepsilon_{\perp}$, the resulting radiation pattern exhibits the symmetric propagation shown in Fig. 4a. In contrast, the radiation pattern in the XZ plane comprises two components ($E_x$ and $E_z$), allowing the field to probe the hyperbolic dispersion. Consequently, a point source positioned above a type-II hyperbolic medium (Z < 0) produces a sharply defined hyperbolic radiating pattern in the XZ planes (Fig. 4b), which is consistent with the isofrequency surface at 6.36 eV (195 nm) in reciprocal space in Fig. 3b. On the other hand, the point source launched at 4.96 eV (250 nm), which is outside of the hyperbolic window, does not show directional propagation as expected from its isofrequency contour (Supplementary Fig. 12). Since HEP supports large momentum, they are difficult to excite via direct free-space incidence in the far field[1]. Thus, using a high-index prism, such as the Otto and Kretschmann configuration under TM-polarized illumination enables efficient coupling to these high-k modes (Supplementary Fig. 13)[63].

This hyperbolic dispersion enables highly directional propagation in the XY and XZ planes for type-I and II, respectively, and leads to strong electric field confinement of HEP in the near field, in contrast to elliptical dispersion materials where energy spreads across all

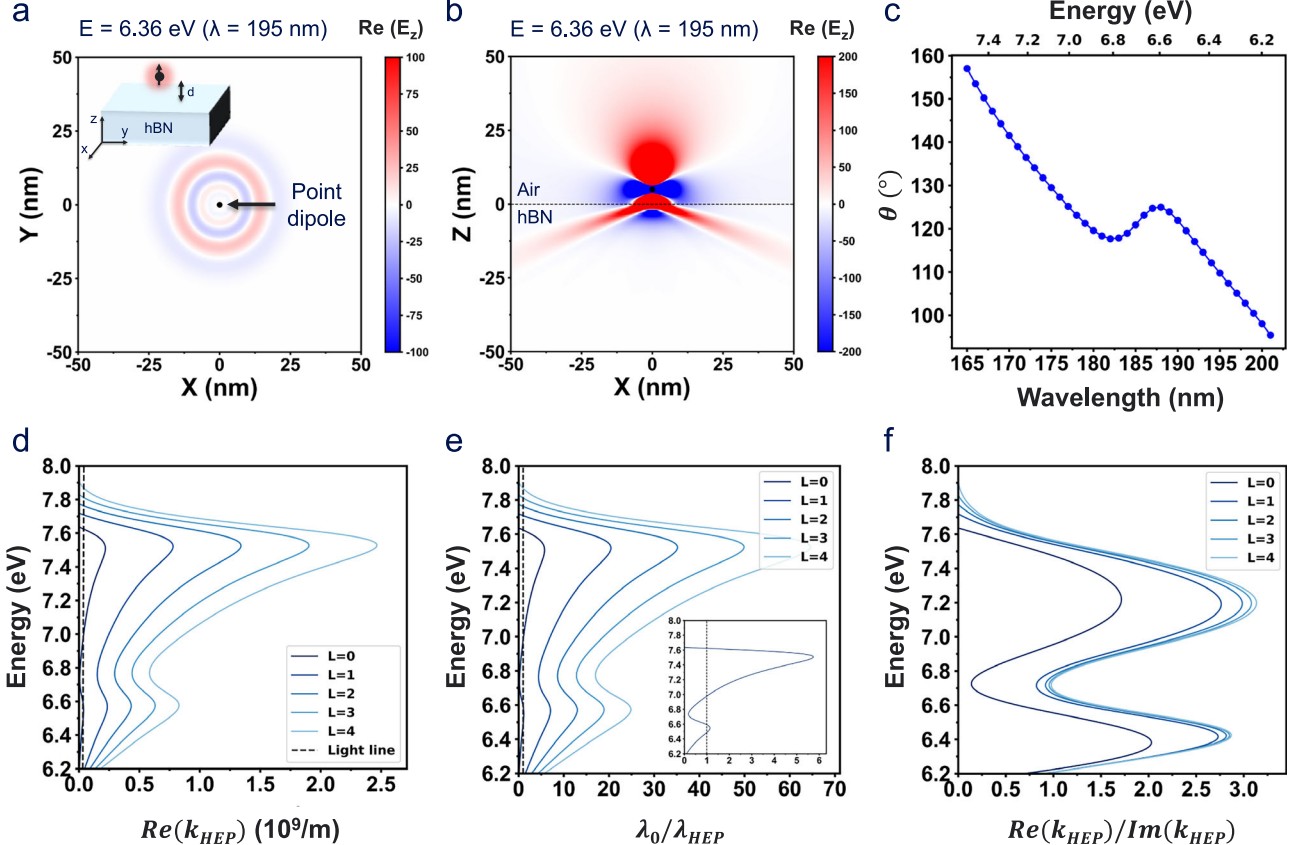

**Fig. 4 | Hyperbolic exciton polaritons (HEP) propagation in hBN slab.** Numerically simulated near-field wave propagation at 6.36 eV (195 nm) using finite-difference time-domain (FDTD) simulations, launched by a point dipole located above the hBN slab (d = 5 nm) and oriented along the Z-axis, is shown in (**a**) the in-plane (XY) direction, revealing isotropic and radial propagation, and **b** the out-of-plane (XZ) direction, exhibiting the highly directional wave propagation in the hBN layer (Z < 0) associated with the hyperbolic dispersion, unlike the air (Z > 0). **c** Wave propagation angle as a function of photon energy (wavelength). **d** The HEP's analytical dispersion relation as a function of an in-plane wavevector up to L = 4 mode, where $d_{hBN}$ is 10 nm, showing the highly confined nature of HEP. Calculated figure of merits: **e** Confinement factor ($\lambda_0/\lambda_{HEP}$, inset shows zoomed-in L = 0 mode) and **f** Propagation loss properties (Re($k_{HEP}$)/Im($k_{HEP}$)) of HEP as a function of incident photon energy, showing highly confined propagation characteristic.

planes[15,64]. In addition, HEP exhibits directional propagation depending on the frequency, and the propagation angle (θ), the angle between the Poynting vector and the z axis, can be approximately calculated by Eq. (2)[34]:

$$\theta(\omega) = \pi/2 - \tan^{-1}(\sqrt{\varepsilon_\perp(\omega)}/i\sqrt{\varepsilon_\parallel(\omega)}) \qquad (2)$$

In this regime, polaritons can only propagate at specific angles determined by the anisotropy of the material. The propagation angle (θ) decreases from 157° to 95° (Δθ ≅ 62°) within the hyperbolic window as photon energy decrease (wavelength increases) (Fig. 4c). Simulated wave propagation in the XZ plane as a function of photon energy exhibits pronounced directionality, in agreement with the calculations shown in Fig. 4c. (Supplementary Fig. 14). We analytically solve the HEP dispersion relation for TM modes, in the thin hBN slab following the general equations for the TM polariton mode given by Eq. (3)[65]:

$$k_{HEP} = \frac{i}{d_{hBN}}\sqrt{\frac{\varepsilon_\perp}{\varepsilon_\parallel}}[2\tan^{-1}\left(i\frac{1}{\sqrt{\varepsilon_\perp\varepsilon_\parallel}}\right) + \pi L] \qquad (3)$$

where $k_{HEP}$ is the x component of the wavenumber, $d_{hBN}$ is the thickness of the hBN slab and L = 0, 1, 2 … are integers indicating the higher order mode. The analytical dispersion relation derived from Eq. (3) clearly reveals the HEP mode, which carries a larger momentum than that of a free-space photon, where the dashed line represents the light line (Fig. 4d). As expected for the hyperbolic polariton, the HEP's dispersion relation exhibits an increasing momentum as mode order increases. In addition, the group velocity ($v_g = \partial\omega/\partial k_{HEP}$), the slope of the dispersion curve, decreases with increasing momentum, reflecting the slow group velocity and strong spatial confinement of higher-order modes[17]. We extract the commonly used figures of merit for the polariton: the confinement factor ($\lambda_0/\lambda_{HEP}$) where $\lambda_0$ is the free-space wavelength, and propagation figure of merit of HEP, defined as Re($k_{HEP}$)/Im($k_{HEP}$)[60,66]. Figure 4e shows the confinement factor depending on the mode order, showing the highly confined nature of HEP with an increase in confinement as the mode order increases. For the L = 0 mode, the confinement factor reaches approximately 5.7, which is comparable to the values observed in exciton-polaritons in WSe₂ (Fig. 4e inset)[67]. In addition, the confinement factor reaches values exceeding 10 for higher-order modes, underscoring the extreme subwavelength confinement enabled by the hyperbolic dispersion in hBN. As shown in Fig. 4f, higher propagation figure of merit values indicates reduced relative propagation losses of HEP. This suggests that, despite their inherently large momentum, HEP can still propagate an appreciable distance (~20 nm) before significant attenuation occurs[54,65]. Furthermore, the value of propagation figure of merit increases with mode order, reflecting improved loss-performance characteristics in higher-order modes. The calculated propagation length along the direction corresponding to the maximum real part of the wavevector is approximately 20 nm

(Supplementary Fig. 15a)[61]. In addition, the calculated HEP lifetime is approximately 0.1 fs, which is comparable to the values for HEP in hBN/WS$_2$/hBN heterostructures[65] (Supplementary Fig. 15b).

In conclusion, from the spectroscopic ellipsometry analysis, we infer the emergence of natural hyperbolic dispersion in hBN arising from strong and anisotropic excitonic resonances in the DUV regime. Using ISE with micron resolution, we extract the complex dielectric functions along both in-plane and out-of-plane directions, revealing a clear type-II hyperbolic window induced by excitonic anisotropy. We associate the highly reflective plateau with hyperbolicity through reflectance spectroscopy, showing good agreement with numerical calculations. Moreover, we suggest that this hyperbolic dispersion supports highly confined polaritonic modes with large momenta, slow group velocities, and enhanced PDOS. Numerical simulations and analytical modeling of the HEP dispersion relation show strong confinement factors and moderate propagation figures of merit. These findings suggest hBN as a candidate for DUV nanophotonics, providing a pathway toward next-generation polaritonic devices for sub-diffraction imaging, DUV lithography, and quantum optical applications.

## Method

### Sample preparation
The exfoliated hBN sample was prepared using two methods, direct transfer and pick-up transfer. For the direct transfer, the bulk hBN was exfoliated using blue tape and then directly dry transferred on top of the target substrate. For the pick-up transfer, mechanically exfoliated hBN flakes were picked up using a dry transfer method[68] with polycarbonate (PC) /polydimethylsiloxane (PDMS) and subsequently transferred onto the target substrate. After the transfer, the samples were immersed in chloroform for 1 hour to dissolve the PC film.

### Imaging spectroscopic ellipsometry
Imaging ellipsometry measurements for determining the dielectric function were performed using the Accurion EP4 system (Park Systems) over the 190–1000 nm spectral range. A 7× objective lens was used for measurements in the DUV regime. Multi-angle incidence measurements were conducted over an angular range of 40° to 70°. The measured mapping data were analyzed using the EP4 model and DataStudio software provided by the EP4 system.

### Raman characterization
Raman spectra were acquired using a Horiba LabRam HR Evolution confocal microscope, with a 600 grooves/mm grating and a 633 nm continuous-wave (CW) laser for excitation. The signal was detected using a CCD detector.

### Reflectance measurements
Reflectance measurements were performed using a Hamamatsu deuterium lamp as the light source. The DUV light was guided into a home-built confocal microscopy set-up, which was constructed using DUV-enhanced aluminum mirrors, a DUV-enhanced reflective objective with a numerical aperture of 0.4, CaF$_2$ lenses, and a CaF$_2$ beam splitter. A 4f optical system was implemented, and a size-tunable aperture was placed at the image plane to spatially filter the reflected signal, thereby isolating the reflectance from the hBN samples while blocking contributions from the underlying substrate. The reflected light was guided into a spectrometer and detected using an electrically cooled Si CCD camera.

### Optical simulations
Electric field profiles in the near field were calculated using FDTD simulations performed with the commercial software Lumerical. Theoretical reflectance and absorbance are computed using the TMM with homemade Python code[57,58]. The refractive index of hBN was determined through imaging ellipsometry, while those of Al$_2$O$_3$[69] and Al[70] were adopted from previously reported literature values.

## Data availability
Relevant data supporting the key findings of this study are available within the article and the Supplementary Information file. All raw data generated during the current study are available from the corresponding authors upon request.

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

## Acknowledgements

D. J., B.C., and J.L. acknowledge support from the Office of Naval Research Young Investigator Award, Metamaterials Program (N00014-23–1-2037). This work was carried out in part at the Singh Center for Nanotechnology, which is supported by the NSF National Nanotechnology Coordinated Infrastructure Program under Grant NNCI-2025608. J.K. acknowledges the support from the Institute for Basic Science (IBS), Korea under Project Code IBS-R014-A1 and the National Research Foundation of Korea grants (NRF-2023R1A2C2007998). N. E. acknowledges partial support from the US Air Force Office of Scientific Research (AFOSR) Multidisciplinary University Research Initiatives (MURI) program with grant # FA9550-21–1-0312 and AFOSR grant # FA9550-23-1-0307.

## Author contributions

D.J. supervised and acquired funding for the project. D.J. and B.C. conceived and designed the experiment. B.C. performed the ellipsometry measurements and conducted the data fitting. B.C. performed the optical simulations and calculations in discussion with D.J., N.E., and J.L. W.C. fit preliminary ellipsometry data. K.Y. fabricated samples. S. J. and H.C. performed the reflectance measurement under the supervision of J.K., B.C., and D.J. wrote the manuscript with inputs from all authors. All authors discussed the results and revised the manuscript.

## Competing interests

The authors declare no competing interests.
