## [Transparent Peer Review file · Nature Communications]

Natural Hyperbolicity of Hexagonal Boron Nitride in the Deep Ultraviolet

Corresponding Author: Professor Deep Jariwala

Version 0:

Reviewer comments:

Reviewer #1

(Remarks to the Author)

I thank the authors for the detailed reply and for tackling all my concerns. I think the current version of the manuscript is perfectly suitable for publication in Nature Communications.

Reviewer #2

(Remarks to the Author)

Natural Hyperbolicity of Hexagonal Boron Nitride in the Deep Ultraviolet
Choi et al.

These comments supplement my previous report.

While the experimental work is of quality and the data is very valuable to the community, the manuscript's narrative fundamentally misrepresents what was actually demonstrated.

In this work, the authors report imaging spectroscopy ellipsometry measurements in the UV range, from which the complex dielectric function can be extracted, along with reflectance spectra. That constitutes the entirety of the experimental evidence. All aspects relating to hyperbolic dispersion are derived from simulation.

Finding that the product of both orthogonal permittivities is negative establishes a necessary condition for hyperbolicity, not proof of the phenomenon. Yet, the authors repeatedly assert, both in the manuscript and in their rebuttal, that their "ellipsometric measurements directly capture [...] the hyperbolic dispersion of hBN". This statement is incorrect. Ellipsometry is a far-field technique that measures bulk dielectric response; it cannot directly access the near-field, high-momentum modes that define hyperbolic behavior.

The treatment of hyperbolicity is theoretical and, while interesting, carries limited impact.

Specific observable consequences must be measured to claim hyperbolic dispersion: measured dispersion, support for high-k modes, directional propagation or open isofrequency curves. Ellipsometry, albeit very powerful, cannot directly capture these effects: they require SNOM, EELS, prisms, gratings, or Raman.

The language is, in my opinion, misleading through the manuscript. The title "Natural Hyperbolicity of h-BN." suggests that the hyperbolicity was measured, not inferred. Same about several sentences in the abstract, the body of the manuscript, and the conclusion.

Furthermore, they argue that the reflective plateau "confirms" hyperbolicity. This conclusion is unwarranted. This plateau arises because of the negative permittivity. The material is metal-like at some frequencies, which is the case for almost all materials, irrespective of anisotropy or hyperbolicity.

The applications are overstated: “Our findings enable light manipulation at the nanoscale in the DUV regime”, “providing a pathway toward next-generation polaritonic devices for sub-diffraction imaging, DUV lithography and quantum applications.”, and “extreme-resolution photolithography”. These claims are not justified by the work presented. The authors’ own calculations report very short propagation lengths and lifetimes, indicating important losses.

Reviewer #3

(Remarks to the Author)

I find the authors’ response to my previous comments highly satisfactory, and I recommend its publication in Nature Communications.

The images or other third party material in this Peer Review File are included in the article’s Creative Commons license, unless indicated otherwise in a credit line to the material. If material is not included in the article’s Creative Commons license and your intended use is not permitted by statutory regulation or exceeds the permitted use, you will need to obtain permission directly from the copyright holder.

REVIEWERS' COMMENTS

Reviewer #1 (Remarks to the Author):

I thank the authors for the detailed reply and for tackling all my concerns. I think the current version of the manuscript is perfectly suitable for publication in Nature Communications.

Our response: We thank the reviewer for their careful evaluation and for recommending the manuscript for publication in its current form.

Reviewer #2 (Remarks to the Author):

Natural Hyperbolicity of Hexagonal Boron Nitride in the Deep Ultraviolet
Choi et al.

These comments supplement my previous report.

While the experimental work is of quality and the data is very valuable to the community, the manuscript's narrative fundamentally misrepresents what was actually demonstrated.

In this work, the authors report imaging spectroscopy ellipsometry measurements in the UV range, from which the complex dielectric function can be extracted, along with reflectance spectra. That constitutes the entirety of the experimental evidence. All aspects relating to hyperbolic dispersion are derived from simulation.

Finding that the product of both orthogonal permittivities is negative establishes a necessary condition for hyperbolicity, not proof of the phenomenon. Yet, the authors repeatedly assert, both in the manuscript and in their rebuttal, that their "ellipsometric measurements directly capture [...] the hyperbolic dispersion of hBN". This statement is incorrect. Ellipsometry is a far-field technique that measures bulk dielectric response; it cannot directly access the near-field, high-momentum modes that define hyperbolic behavior.

The treatment of hyperbolicity is theoretical and, while interesting, carries limited impact.

Specific observable consequences must be measured to claim hyperbolic dispersion: measured dispersion, support for high-k modes, directional propagation or open isofrequency curves. Ellipsometry, albeit very powerful, cannot directly capture these effects: they require SNOM, EELS, prisms, gratings, or Raman.

The language is, in my opinion, misleading through the manuscript. The title "Natural Hyperbolicity of h-BN." suggests that the hyperbolicity was measured, not inferred. Same about several sentences in the abstract, the body of the manuscript, and the conclusion.

Furthermore, they argue that the reflective plateau "confirms" hyperbolicity. This conclusion is unwarranted. This plateau arises because of the negative permittivity. The material is metal-like at some frequencies, which is the case for almost all materials, irrespective of anisotropy or hyperbolicity.

The applications are overstated: “Our findings enable light manipulation at the nanoscale in the DUV regime”, “providing a pathway toward next-generation polaritonic devices for sub-diffraction imaging, DUV lithography and quantum applications.”, and “extreme-resolution photolithography”. These claims are not justified by the work presented. The authors’ own calculations report very short propagation lengths 3λ and lifetimes, indicating important losses.

Our response: We sincerely thank the reviewer for this thorough and constructive comment. We agree that imaging spectroscopic ellipsometry (ISE) probes the far-field bulk dielectric response and does not directly access the near-field, high-momentum modes. Owing to the high-energy (DUV) regime considered here, conventional near-field techniques are difficult to implement, making direct experimental observation of these hyperbolic behaviors, such as support for high-k modes, directional propagation, or open isofrequency curves, particularly challenging. Such measurements would likely require alternative approaches, such as EELS, which are beyond the scope of the present work. We note that in the literature, hyperbolic dispersion is commonly defined by the opposite signs of the principal components of the permittivity tensor [Science 336, 205 (2012); Nat. Photonics 7, 948 (2013); Nat. Photonics 9, 214 (2015); Phys. Rev. Lett. 121, 127401 (2018)]. Accordingly, observing opposite signs of the permittivity tensor components is widely regarded as strong evidence for the existence of a hyperbolic dispersion regime. In this sense, our measurements establish the hyperbolic dispersion regime, while the existence of hyperbolic exciton polaritons (HEP) and related phenomena is inferred through theory and simulations.

Accordingly, we have carefully revised the manuscript to eliminate statements implying a direct experimental observation of hyperbolicity. Throughout the abstract, main text, and conclusions, we now consistently present hyperbolic behavior as a theoretically inferred consequence of the experimentally extracted dielectric tensor, rather than as a directly measured phenomenon. We have rephrased all language suggesting that ellipsometry “directly captures” hyperbolic dispersion. We have also revised the discussion of the reflectance plateau to avoid attributing it as confirmation of hyperbolicity. Furthermore, we have moderated the statements regarding applications.

We believe these revisions now present a more accurate and balanced description of what is experimentally demonstrated and what is theoretically inferred, in line with the reviewer’s concerns.

Change to the manuscript:

"Here, based on experimental evidence and theoretical fitting estimates to the experimental data, we suggest the presence of natural hyperbolic dispersion in

hexagonal boron nitride (hBN) in the deep-ultraviolet (DUV) regime,... " (page 1, lines 26-27)

"...revealing a **potential** type-II hyperbolic window in the DUV regime." (page 1, line 32)

"Our findings **suggest** hBN as a **promising** platform for nanophotonic applications..." (page 1, line 34)

"hBN shows pronounced exciton oscillator strength in the in-plane direction compared to the out-of-plane direction, **suggesting the presence of a** hyperbolic window in the DUV regime." (page 3, lines 5-6)

"**Our findings would enable** light manipulation at the nanoscale in the DUV regime..." (page 3, line 10)

"Based on the dispersion relation, Figure 3b shows the clear hyperbolic **isofrequency surface** of hBN at 6.36 eV (195 nm) for each lossy (red and blue) and lossless (green) case, ..." (page 9, lines 12-13)

"The anisotropic excitonic resonance **suggests the possibility of a** hyperbolic dispersion, .." (page 9, lines 15-16)

"We further investigate the highly confined HEP **associated with the** hyperbolic dispersion." (page 10, lines 14)

"In conclusion, **from the spectroscopic ellipsometry analysis, we infer the emergence of** natural hyperbolic dispersion in hBN arising from strong and anisotropic excitonic resonances in the DUV regime." (page 12, lines 3-4)

"**We associate the highly reflective plateau with hyperbolicity through reflectance spectroscopy,** showing good agreement with numerical calculations. Moreover, **we suggest that** this hyperbolic dispersion supports highly confined polaritonic modes with large momenta, slow group velocities, and enhanced PDOS." (page 12, lines 9-10)

"~~We confirm this through~~ Numerical simulations and analytical modeling of the HEP dispersion relation show strong confinement factors and moderate propagation figures of merit." (page 12, lines 11-12)

"These findings **suggest** hBN as a **promising** candidate for DUV nanophotonics..." (page 12, lines 13)

Reviewer #3 (Remarks to the Author):

I find the authors' response to my previous comments highly satisfactory, and I recommend its publication in Nature Communications.

Our response: We thank the reviewer for their careful evaluation and for finding our responses satisfactory and recommending the manuscript for publication in its current form.